# BALANCED AND DETERMINISTIC WEIGHT-SHARING HELPS NETWORK PERFORMANCE

## ABSTRACT

Weight-sharing plays a significant role in the success of many deep neural networks, by increasing memory efficiency and incorporating useful inductive priors about the problem into the network. But understanding how weight-sharing can be used effectively in general is a topic that has not been studied extensively. Chen et al. (2015) proposed HashedNets, which augments a multi-layer perceptron with a hash table, as a method for neural network compression. We generalize this method into a framework (ArbNets) that allows for efficient arbitrary weight-sharing, and use it to study the role of weight-sharing in neural networks. We show that common neural networks can be expressed as ArbNets with different hash functions. We also present two novel hash functions, the Dirichlet hash and the Neighborhood hash, and use them to demonstrate experimentally that balanced and deterministic weight-sharing helps with the performance of a neural network.

## 1 INTRODUCTION

Most deep neural network architectures can be built using a combination of three primitive networks: the multi-layer perceptron (MLP), the convolutional neural network (CNN), and the recurrent neural network (RNN). These three networks differ in terms of where and how the weight-sharing takes place. We know that the weight-sharing structure is important, and in some cases essential, to the success of the neural network at a particular machine learning task.

For example, a convolutional layer can be thought of as a sliding window algorithm that shares the same weights applied across different local segments in the input. This is useful for learning translation-invariant representations. Zhang et al. (2017) showed that on a simple ten-class image classification problem like CIFAR10, applying a pre-processing step with $32,000$ random convolutional filters boosted test accuracy from $54\%$ to $83\%$ using an SVM with a vanilla Gaussian kernel. Additionally, although the ImageNet challenge only started in 2010, from 2012 onwards, all the winning models have been CNNs. This suggests the importance of convolutational layers for the task of image classification. We show later on that balanced and deterministic weight-sharing helps network performance, and indeed, the weights in convolutional layers are shared in a balanced and deterministic fashion.

We also know that tying the weights of encoder and decoder networks can be helpful. In an autoencoder with one hidden layer and no non-linearities, tying the weights of the encoder and the decoder achieves the same effect as Principal Components Analysis (Roweis, 1997). In language modeling tasks, tying the weights of the encoder and decoder for the word embeddings also results in increased performance as well as a reduction in the number of parameters used (Inan et al., 2017).

Developing general intuitions about where and how weight-sharing can be leveraged effectively is going to be very useful for the machine learning practitioner. Understanding the role of weight-sharing in a neural network from a quantitative perspective might also potentially lead us to discover novel neural network architectures. This paper is a first step towards understanding how weight-sharing affects the performance of a neural network.

We make four main contributions:

- We propose a general weight-sharing framework called ArbNet that can be plugged into any existing neural network and enables efficient arbitrary weight-sharing between its parameters. (Section 1.1)

- We show that deep networks can be formulated as ArbNets, and argue that the problem of studying weight-sharing in neural networks can be reduced to the problem of studying properties of the associated hash functions. (Section 2.4)

- We show that balanced weight-sharing increases network performance. (Section 5.1)

- We show that making an ArbNet hash function, which controls the weight-sharing, more deterministic increases network performance, but less so when it is sparse. (Section 5.2)

## 1.1 ARBNET

ArbNets are neural networks augmented with a hash table to allow for arbitrary weight-sharing. We can label every weight in a given neural network with a unique identifier, and each identifier maps to an entry in the hash table by computing a given hash function prior to the start of training. On the forward and backward passes, the network retrieves and updates weights respectively in the hash table using the identifiers. A hash collision between two different identifiers would then imply weight-sharing between two weights. This mechanism of forcing hard weight-sharing is also known as the 'hashing trick' in some machine learning literature.

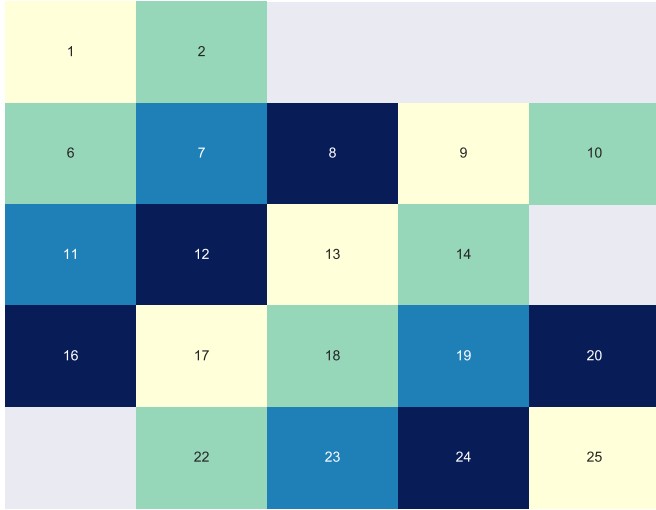

Figure 1: A visualization of a one-layer MLP ArbNet with the identifiers numbered sequentially. The weight matrix is of size $5$ by $5$ with a modulus hash into a table of size $4$. Weights with the same color are tied to each other. The gray represents the absence of a weight, which occurs frequently if the network is sparse.

A simple example of a hash function is the modulus hash:

$$w_i = table_{i \bmod n} \tag{1}$$

where the weight $w_i$ with identifier $i$ maps to the $(i \bmod n)$th entry of a hash table of size $n$.

An ArbNet is an efficient mechanism of forcing weight-sharing between any two arbitrarily selected weights, since the only overhead involves memory occupied by the hash table and the identifiers, and compute time involved in initializing the hash table.

## 1.2 How the Hash Function Affects Network Performance

As the load factor of the hash table goes up, or equivalently as the ratio of the size of the hash table relative to the size of the network goes down, the performance of the neural network goes down. This was demonstrated in Chen et al. (2015). While the load factor is a variable controlling the capacity of the network, it is not necessarily the most important factor in determining network performance. A convolutional layer has a much higher load factor than a fully connected layer, and yet it is much more effective at increasing network performance in a range of tasks, most notably image classification.

There are at least two other basic questions we can ask:

- How does the balance of the hash table affect performance?
    - The balance of the hash table indicates the evenness of the weight sharing. We give a more precise definition in terms of Shannon entropy in the EXPERIMENTAL SETUP section, but intuitively, a perfectly balanced weight sharing scheme accesses each entry in the hash table the same number of times, while an unbalanced one would tend to favor using some entries more than others.
- How does noise in the hash function affect performance?
    - For a fixed identifier scheme, if the hash function is a deterministic operation, it will map to a fixed entry in the hash table. If it is a noisy operation, we cannot predict a priori which entry it would map into.
    - We do not have a rigorous notion for 'noise', but we demonstrate in the EXPERIMENTAL SETUP section an appropriate hash function whose parameter can be tuned to tweak the amount of noise.

We are interested in the answers to these question across different levels of sparsity, since as in the case of a convolutional layer, this might influence the effect of the variable we are studying on the performance of the neural network. We perform experiments on two image classification tasks, MNIST and CIFAR10, and demonstrate that balance helps while noise hurts neural network performance. MNIST is a simpler task than CIFAR10, and the two tasks show the difference, if any, when the neural network model has enough capacity to capture the complexity of the data versus when it does not.

## 2 Common Neural Networks are MLP ArbNets

The hash function associated with an MLP ArbNet is an exact specification of the weight-sharing patterns in the network, since an ordinary MLP does not share any weights.

### 2.1 Multi-layer Perceptrons

An MLP consists of repeated applications of fully connected layers:

$$y_i = \sigma_i(W_i x_i + b_i) \tag{2}$$

at the $i$th layer of the network, where $\sigma_i$ is an activation function, $W_i$ a weight matrix, $b_i$ a bias vector, $x_i$ the input vector, and $y_i$ the output vector. None of the weights in any of the $W_i$ are shared, so we can consider an MLP in the ArbNet framework as being augmented with identity as the hash function.

### 2.2 Convolutional Neural Networks

A CNN consists of repeated applications of convolutional layers, which in the 2D case, resembles an equation like the following:

$$Y_i^{(j,k)} = \sigma_i(\sum_m \sum_n W_i^{(m,n)} X_i^{(j-m,k-n)} + B_i^{(j,k)}) \tag{3}$$

at the $i$th layer of the network, where the superscripts index the matrix, $\sigma_i$ is an activation function (includes pooling operations), $W_i$ a weight matrix of size $m$ by $n$, $X_i$ the input matrix of size $a$ by $b$, $B_i$ a bias matrix and $Y_i$ the output matrix. The above equation produces one feature map. To produce $l$ feature maps, we would have $l$ different $W_i$ and $B_i$, and stack the $l$ resultant $Y_i$ together. Notice that equation 3 can be rewritten in the form of equation 2:

$$y_i = \sigma_i(W'_i x_i + b_i) \tag{4}$$

where the weight matrix $W'_i$ is given by the equation:

$$W'^{(u,v)}_i = \begin{cases} W_i^{(\lfloor (u+v)/b \rfloor, (u+v) \bmod b)} & \text{if the indices are defined for } W_i \\ 0 & \text{otherwise} \end{cases} \tag{5}$$

$W'_i$ has the form of a sparse Toeplitz matrix, and we can write a CNN as an MLP ArbNet with a hash function corresponding to equation 5. Convolutions where the stride or dilation is more than one have not been presented for ease of explanation but analogous results follow.

## 2.3 RECURRENT NEURAL NETWORKS

An RNN consists of recurrent layers, which takes a similar form as equation 2, except that $W_i = W_j$ and $B_i = B_j$ for $i \neq j$, i.e. the same weights are being shared across layers. This is equivalent to an MLP ArbNet where we number all the weights sequentially and the hash function is a modulus hash (equation 1) with $n$ the size of each layer.

## 2.4 GENERAL NETWORKS

We have shown above that MLPs, CNNs, and RNNs can be written as MLP ArbNets associated with different hash functions. Since deep networks are built using a combination of these three primitive networks, it follows that deep networks can be expressed as MLP ArbNets. This shows the generality of the ArbNet framework.

Fully connected layers do not share any weights, while convolutional layers share weights within a layer in a very specific pattern resulting in sparse Toeplitz matrices when flattened out, and recurrent layers share the exact same weights across layers. The design space of potential neural networks is extremely big, and one could conceive of effective weight-sharing strategies that deviate from these three standard patterns of weight-sharing.

In general, since any neural network, not just MLPs, can be augmented with a hash table, ArbNets are a powerful mechanism for studying weight-sharing in neural networks. The problem of studying weight-sharing in neural networks can then be reduced to the problem of studying the properties of the associated hash functions. As a proof of concept, we converted a DenseNet (Huang et al., 2017) with $769,000$ weights into an ArbNet with a uniform hash (equation 6), which is a uniform distribution on the entries of the hash table. On CIFAR10 without any data augmentation, the original DenseNet performed at $95\%$ accuracy, while the ArbNet version performed at $85\%$ accuracy using a hash table of size $10,000$ (76x reduction in parameters) and $92\%$ accuracy using a hash table of size $100,000$ (7.6x reduction in parameters). It is remarkable that such arbitrary weight-sharing paired with the standard stochastic gradient descent (SGD) training procedure can still result in models with high accuracy.

$$w_i = table_{Uniform(n)} \tag{6}$$

We describe ArbNets above as involving a single hash table and a single hash function, but we can also use multiple hash tables and multiple hash functions in general. For example, we can choose to do layer-wise hashing, where each individual layer in a feed-forward network has an associated hash table and function, which will be equivalent to some network-wise hashing scheme with only a single hash table.

# 3 RELATED WORK

Chen et al. (2015) proposed HashedNets for neural network compression, which is an MLP ArbNet where the hash function is computed layer-wise using `xxHash` prior to the start of training. Han et al. (2016) also made use of the same layer-wise hashing strategy for the purposes of network compression, but hashed according to clusters found by a K-means algorithm run on the weights of a trained network. Our work generalizes this technique, and uses it as an experimental tool to study the role of weight-sharing.

Besides hard weight-sharing, it is also possible to do soft weight-sharing, where two different weights in a network are not forced to be equal, but are related to each other. Nowlan & Hinton (1992) implemented a soft weight-sharing strategy for the purposes of regularization where the weights are drawn from a Gaussian mixture model. Ullrich et al. (2017) also used Gaussian mixture models as soft weight-sharing for doing network compression.

Another soft weight-sharing strategy called HyperNetworks (Ha et al., 2017) involves using a LSTM controller as a meta-learning algorithm to generate the weights of another network.

# 4 EXPERIMENTAL SETUP

In this paper, we limit our attention to studying certain properties of MLP ArbNets as tested on the MNIST and CIFAR10 image classification tasks. Our aim is not to best any existing benchmarks, but to show the differences in test accuracy as a result of changing various properties of the hash function associated with the MLP ArbNet.

## 4.1 BALANCE OF THE HASH TABLE

The balance of the hash table can be measured by Shannon entropy:

$$H = -\sum_i p_i \log p_i \tag{7}$$

where $p_i$ is the probability that the $i$th table entry will be used on a forward pass in the network. We propose to control this with a Dirichlet hash, which involves sampling from a symmetric Dirichlet distribution and using the output as the parameters of a multinomial distribution which we will use as the hash function. The symmetric Dirichlet distribution has the following probability density function:

$$P(X) = \frac{\Gamma(\alpha N)}{\Gamma(\alpha)^N} \prod_{i=1}^{N} x_i^{\alpha-1} \tag{8}$$

where the $x_i$ lie on the $N-1$ simplex. The Dirichlet hash can be given by the following function:

$$w_i = table_{Multinomial_\alpha(n)} \tag{9}$$

A high $\alpha$ leads to a balanced distribution (high Shannon entropy), and a low $\alpha$ leads to an unbalanced distribution (low Shannon entropy). The limiting case of $\alpha \to \infty$ results in a uniform distribution, which has maximum Shannon entropy. See Figure 2 for a visualization of the effects of $\alpha$ on a hash table with 1000 entries.

## 4.2 NOISE IN THE HASH FUNCTION

A modulus hash and a uniform hash both have the property that the expected load of all the entries in the hash table is the same. Hence, in expectation, both of them will be balanced the same way, i.e. have the same expected Shannon entropy. But the former is entirely deterministic while the latter is entirely random. In this case, it is interesting to think about the effects of this source of noise, if any, on the performance of the neural network. We propose to investigate this with a

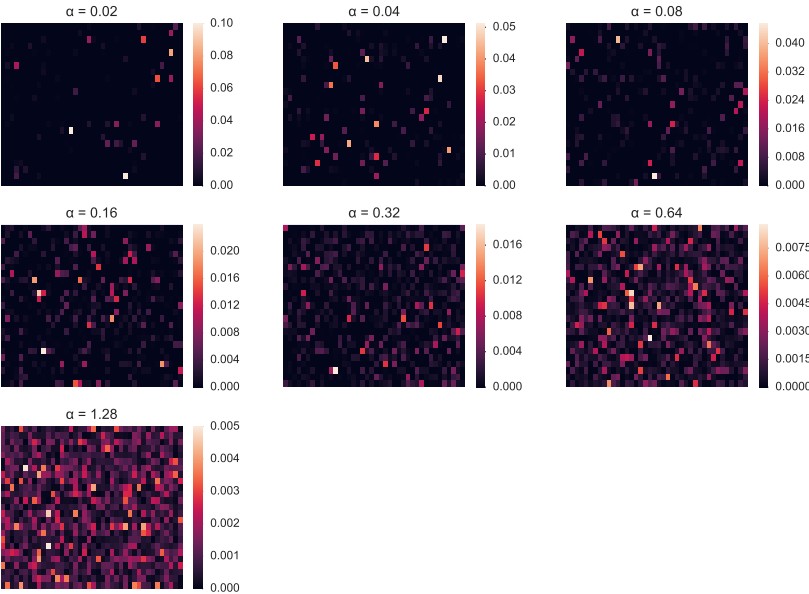

Figure 2: Heatmap of multinomial parameters drawn from different values of $\alpha$

Neighborhood hash, which involves the composition of a modulus hash and a uniform distribution around a specified $radius$. This is given by the following hash function:

$$w_i = table_{(i+Uniform([-radius,\ radius]))\ \mathrm{mod}\ n} \qquad (10)$$

When the $radius$ is 0, the Neighborhood hash reduces to the modulus hash, and when the radius is at least half the size of the hash table, it reduces to the uniform hash. Controlling the radius thus allows us to control the intuitive notion of 'noise' in the specific setting where the expected load of all the table entries is the same.

### 4.3 NETWORK SPECIFICATION

On MNIST, our ArbNet is a three layer MLP (200-ELU-BN-200-ELU-BN-10-ELU-BN) with exponential linear units (Clevert et al., 2016) and batch normalization (Ioffe & Szegedy, 2015).

On CIFAR10, our ArbNet is a six layer MLP (2000-ELU-BN-2000-ELU-BN-2000-ELU-BN-2000-ELU-BN-2000-ELU-BN-10-ELU-BN) with exponential linear units and batch normalization.

We trained both networks using SGD with learning rate 0.1 and momentum 0.9, and a learning rate scheduler that reduces the learning rate 10x every four epochs if there is no improvement in training accuracy. No validation set was used.

For both datasets, we used a layer-wise hash function, where each layer with $n$ weights was numbered sequentially from 1 to $n$ and then hashed into a hash table (size 1000 for MNIST and size 10,000 for CIFAR10). To be clear, there are three hash tables in the ArbNet for MNIST and six for CIFAR10. We did the same experiment with a network-wise hash function where all the weights in the entire network with $n$ weights were numbered sequentially from 1 to $n$ and then hashed into a single hash table of size (1000 for MNIST, 10,000 for CIFAR10)∗(number of layers), but similar trends were found, so we will omit mentioning those results. We also tried experimenting with not hashing the bias weights in the MLPs, and similar results were found. It is interesting that layer-wise and network-wise hashes do not produce different results in the case of these two datasets and networks, but we suspect that this is not true in general, since the resultant hash functions are in fact different.

# 5 RESULTS AND DISCUSSION

## 5.1 DIRICHLET HASH

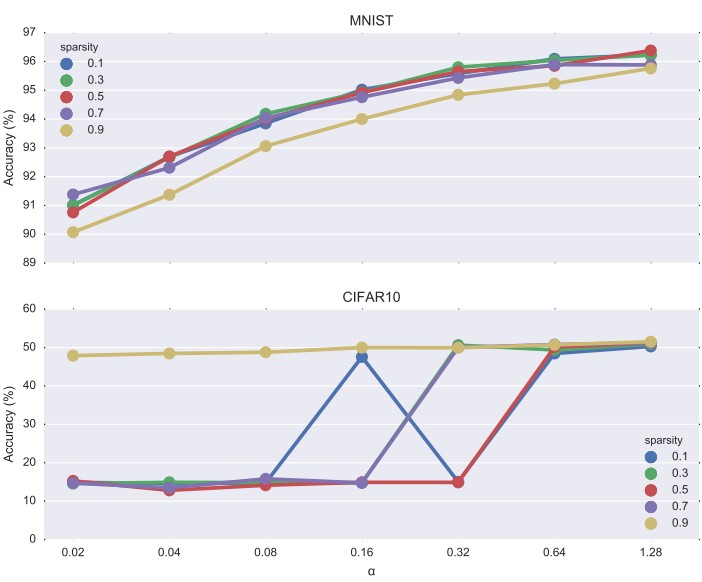

Figure 3: Effect of $\alpha$ (Balance) in Dirichlet hash on network accuracy across different levels of sparsity

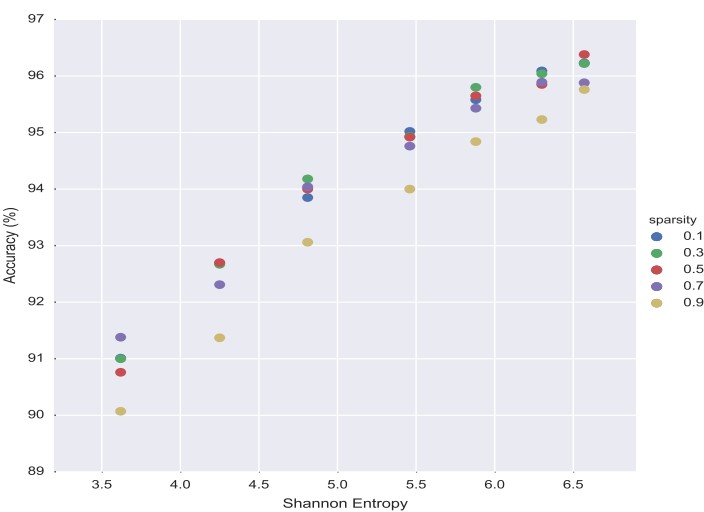

Figure 4: Effect of Shannon entropy (Balance) in Dirichlet hash on network accuracy across different levels of sparsity

We observe in Figure 3 that on the MNIST dataset, increasing $\alpha$ has a direct positive effect on test accuracy, across different levels of sparsity. On the CIFAR10 dataset, when the weights are sparse, increasing $\alpha$ has a small positive effect, but at lower levels of sparsity, it has a huge positive effect. This finding seems to indicate that it is more likely for SGD to get stuck in local minima when the weights are both non-sparse and shared unevenly.

We can conclude that balance helps with network performance, but it is unclear if it brings diminishing returns. Re-plotting the MNIST graph in Figure 3 with the x-axis replaced with Shannon Entropy (equation 7) instead of $\alpha$ in Figure 4 gives us a better sense of scale. Note that in this case, a uniform distribution on 1000 entries would have a Shannon entropy of 6.91. The results shown by Figure 4 suggest a linear trend at high sparsity and a concave trend at low sparsity, but more evidence is required to come to a conclusion.

## 5.2 NEIGHBORHOOD HASH

The trends in Figure 5 are noisier, but it seems like an increase in $radius$ has the overall effect of diminishing test accuracy. On MNIST, we notice that higher levels of sparsity result in a smaller drop in accuracy. The same effect seems to be present but not as pronounced in CIFAR10, where we note an outlier in the case of sparsity 0.1, $radius$ 0. We hypothesize that this effect occurs because the increase in noise leads to the increased probability of two geometrically distant weights in the network being forced to share the same weights. This is undesirable in the task of image classification, where local weight-sharing is proven to be advantageous, and perhaps essential to the task. When the network is sparse, the positive effect of local weight-sharing is not prominent, and hence the noise does not affect network performance as much.

Thus, we can conclude that making the ArbNet hash more deterministic (equivalently, less noisy) boosts network performance, but less so when it is sparse.

We notice that convolutional layers, when written as an MLP ArbNet as in equation 5, have a hash function that is both balanced (all the weights are used with the same probability) and deterministic (the hash function does not have any noise in it). This helps to explain the role weight-sharing plays in the success of convolutional neural networks.

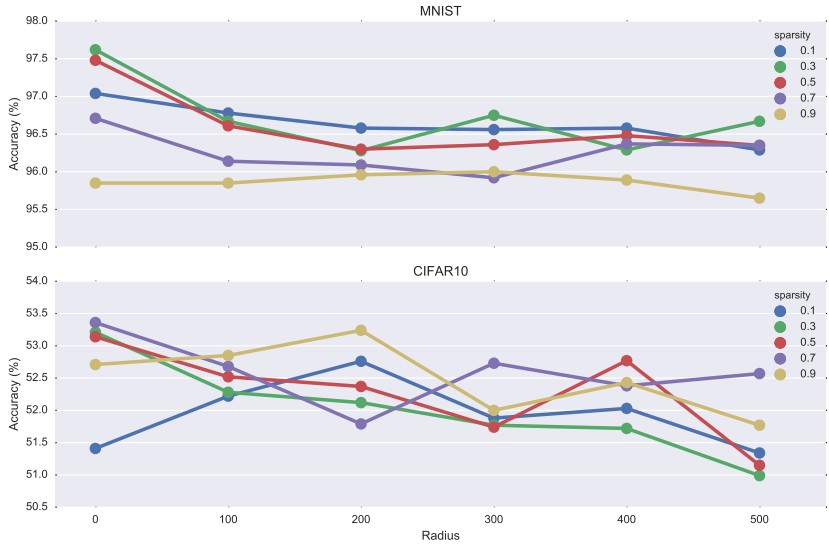

Figure 5: Effect of $radius$ (Noise) in Neighborhood hash on network accuracy across different levels of sparsity

## 6 CONCLUSION

Weight-sharing is very important to the success of deep neural networks, and is a worthy topic of study in its own right. We proposed the use of ArbNets as a general framework under which weight-sharing can be studied, and investigated experimentally, for the first time, how balance and noise affects neural network performance in the specific case of an MLP ArbNet and two image classification datasets. We hope to carry out more extensive experimental and theoretical research on the role of weight-sharing in deep networks, and hope that others will do so too.

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
