# OpenReview forum: "Balanced and Deterministic Weight-sharing Helps Network Performance"
_ICLR.cc/2018/Conference — Reject_

### Official Review · AnonReviewer3 · 2017-11-27
**The manuscript contains few insights**

**Rating:** 4
**Confidence:** 4

**Review:**

The manuscript advocates to study the weight sharing in a more systematic way by proposing ArbNets which defines the weight sharing function as a hash function. In this framework, any existing neural network architectures, including CNN and RNN, could be incorporated into ArbNets.

The manuscript is not well written. There are multiple grammar errors and typos. Content-wise, it is already well known that CNN and RNN can be expressed as general MLP with weight sharing. The introduction of ArbNets does not bring much value or insight to this area. So it seems that most content before experimental section is common sense.

In the experimental section, it is interesting to see how different hash function with different level of entropy can affect the performance of neural nets. However, this single observation cannot enrich the whole manuscript. Two questions:
(1) What is the definition of sparsity here, and how is it controlled?
(2) There seems to be a step change in Figure 3. All the results are either between 10 to 20, or near 50. And the blue line goes up and down. Is this expected?

---

### Official Review · AnonReviewer1 · 2017-11-27
**A framework for studying weight sharing**

**Rating:** 4
**Confidence:** 4

**Review:**

This paper proposes a general framework for studying weight sharing in neural networks. They further suggest two hash functions and study the role of different properties of these hash functions in the performance.

The paper is well-written and clear. It is a follow-up on Chen et al. (2015) which introduced HashedNets. Therefore, the idea of using hash functions is not novel. This paper suggests a framework to study different hash functions. However, the experimental results do not seem adequate to validate this framework. One issue here is lack of a baseline for performance comparison. Otherwise, the significance of the results is not clear.

---

### Official Review · AnonReviewer4 · 2017-12-15
**limited novelty**

**Rating:** 4
**Confidence:** 4

**Review:**

This paper has limited novelty, the ideas has been previously proposed in HashedNet and Deep Compression. The experimental section is week, with only mnist and cifar results it's not convincing to the community whether this method is general.

---

### Decision · Program_Chairs · 2018-01-29
**ICLR 2018 Conference Acceptance Decision**

**Decision:**

Reject

**Comment:**

An empirical study of weight sharing for neural networks is interesting, but all of the reviewers found the experiments insufficient without enough baseline comparisons.